# Fabrication of a Novel 3D Extrusion Bioink Containing Processed Human Articular Cartilage Matrix for Cartilage Tissue Engineering

**DOI:** 10.3390/bioengineering11040329

**Published:** 2024-03-28

**Authors:** Alexandra Hunter Aitchison, Nicholas B. Allen, Isabel R. Shaffrey, Conor N. O’Neill, Bijan Abar, Albert T. Anastasio, Samuel B. Adams

**Affiliations:** 1Department of Orthopaedic Surgery, Duke University Health System, Durham, NC 27710, USA; nba8@duke.edu (N.B.A.); izzy.shaffrey@duke.edu (I.R.S.); conor.n.oneill@duke.edu (C.N.O.); bijan.abar@duke.edu (B.A.); albert.anastasio@duke.edu (A.T.A.); 2Department of Mechanical Engineering, Duke University, Durham, NC 27710, USA

**Keywords:** 3D printing, bioprinting, additive manufacturing, chondrocytes, proliferation, differentiation, *COL2A1*, *SOX9*, decellularized matrix

## Abstract

Cartilage damage presents a significant clinical challenge due to its intrinsic avascular nature which limits self-repair. Addressing this, our study focuses on an alginate-based bioink, integrating human articular cartilage, for cartilage tissue engineering. This novel bioink was formulated by encapsulating C20A4 human articular chondrocytes in sodium alginate, polyvinyl alcohol, gum arabic, and cartilage extracellular matrix powder sourced from allograft femoral condyle shavings. Using a 3D bioprinter, constructs were biofabricated and cross-linked, followed by culture in standard medium. Evaluations were conducted on cellular viability and gene expression at various stages. Results indicated that the printed constructs maintained a porous structure conducive to cell growth. Cellular viability was 87% post printing, which decreased to 76% after seven days, and significantly recovered to 86% by day 14. There was also a notable upregulation of chondrogenic genes, COL2A1 (*p* = 0.008) and SOX9 (*p* = 0.021), suggesting an enhancement in cartilage formation. This study concludes that the innovative bioink shows promise for cartilage regeneration, demonstrating substantial viability and gene expression conducive to repair and suggesting its potential for future therapeutic applications in cartilage repair.

## 1. Introduction

Articular cartilage damage is a prevalent issue that often leads to progressive joint degeneration and development of osteoarthritis, profoundly impacting patients’ function, and quality of life [1]. Osteoarthritis affects around 32 million adults in the United States, a figure expected to increase with the aging population [2]. Conventional treatments, often limited to joint fusion or replacement in late-stage degeneration are costly and provide only temporary symptom relief without restoring the original function of the tissue [3,4]. Therefore, there is significant interest in developing therapies to restore native cartilage integrity and mitigate subsequent cartilage degeneration.

Articular cartilage, a hyaline cartilage comprised of a dense extracellular matrix, chondrocytes, and type II collagen fibers, plays a crucial role in joint movement by reducing friction and absorbing shock [5,6]. Due to its avascular, aneural, and alymphatic nature, articular cartilage has poor capacity for regeneration after damage or wear [7,8]. Tradition interventions for cartilage repair, including osteochondral autograft, allograft, and autologous chondrocyte implantation have demonstrated some success, but face immense challenges such as immune rejection, donor-site morbidity, and limited long-term survival associated with these approaches [9,10]. Osteochondral allografts, commonly used for larger defects that are not amenable to autograft or chondrocyte implantation, are also not readily available, and patients may be required to wait months prior to finding an appropriate match [11].

While these treatments may remain the standard for addressing the mechanical consequences of osteoarthritis, they do not address the underlying biology of cartilage repair. In contrast, emerging research has begun to explore novel materials with potential therapeutic effects on arthritis-related pathologies. For instance, studies on unique materials such as chicken embryo tissue hydrolyzate, illustrate the creative avenues being pursued in the broader research of arthritis treatment [12]. These studies represent a departure from traditional symptom management towards targeting the disease processes, even though their direct application to osteoarthritis remains to be explored.

Within this context of innovation, tissue engineering, particularly bioprinting, offers a promising solution by constructing three-dimensional (3D) scaffolds that imitate the complex architecture of native tissues. Mesenchymal stem cell (MSC) impregnated hydrogels are often used for both articular and fibro-cartilage tissue engineering [13]. This technology leverages a bioink composed of hydrogels, biomaterials, and cells to foster an environment conducive to cell infiltration and proliferation [14,15]. As a result, this technology has immense utility as a therapeutic agent for cartilage degeneration. However, due to lower levels of mechanical integrity when faced with introduction into heavy-load-bearing environments, the efficacy of bioprinted cell-laden scaffolds largely depends on the bioink’s formulation, which must be resilient to printing stresses and compatible with the target tissue [16,17].

Cartilage scaffolds are fabricated using natural biomaterials, synthetic materials such as polycaprolactone (PCL) or other polymers, or a combination of both, each with their own benefits and limitations. Synthetic scaffolds tend to provide greater strength and structural support for the artificial tissue, while natural scaffolds have greater cell compatibility and can enhance cell proliferation [18]. Natural hydrogels have been widely researched for their biocompatibility and supportive properties for cell proliferation, despite their mechanical limitations [19,20,21,22]. Conversely, synthetic scaffolds such as PCL offer improved mechanical strength, but often lack the biological cues necessary to induce cell integration, proliferation, and maturation [23]. While a variety of natural biomaterials have been explored for these purposes, such as bacterial cellulose for its mechanical robustness and chitosan for its bioactive properties, the focus of this study is on the application of alginate as the scaffold biomaterial [24,25,26]. Alginate, known for its rapid cross-linking capabilities and cell-carrying capacity, has shown potential in cartilage tissue engineering, particularly when augmented with other biomaterials to enhance its biological properties [13,27,28,29,30,31,32].

Biomaterials that mimic the cellular microenvironment of target tissues, like decellularized tissue-specific extracellular matrix have gained attention for their ability to replicate the native tissue structure and promote stem cell differentiation and growth [33]. The implementation of decellularized matrices in scaffolds is increasingly explored, with promising results in replicating tissue architecture and inducing native cell proliferation [22,33,34,35]. However, limited studies have explored decellularized cartilage extracellular matrix in the specific context of 3D bioprinted scaffolds for cartilage-tissue-specific engineering, and even fewer use human-derived cartilage as the additive [36].

This study introduces a novel alginate-based bioink infused with human decellularized articular cartilage matrix, aiming to assess its viability for a 3D bioprinting approach to cartilage tissue engineering. We hypothesized that this chondrocyte-laden bioink could suitably endure bioprinting, replicate porous construct morphology, and encourage chondrocyte proliferation and differentiation.

The integration of decellularized matrices, particularly from human-derived cartilage, into bioinks represents a significant advancement in the field of tissue engineering. This approach not only provides a scaffold that is structurally similar to native tissue but also contains biochemical cues that are essential for guiding cell behavior. The use of human-derived cartilage matrix in this study is particularly noteworthy, as it may offer a more physiologically relevant environment for chondrocytes compared to animal-derived or synthetic matrices. This could potentially lead to better integration and functionality of the engineered tissue in human applications.

Furthermore, the focus on 3D bioprinting technology in this study is timely and relevant. Three-dimensional bioprinting has emerged as a powerful tool in medical applications of tissue engineering, allowing for precise control over scaffold architecture and cell distribution. This level of control is crucial in replicating the complex structure of articular cartilage, which is essential for its function in joint movement. The use of an alginate-based bioink, known for its biocompatibility and ease of printing, further enhances the potential of this approach. By combining the advantages of alginate with the biological relevance of human-derived cartilage matrix, this study sets the stage for developing more effective and clinically relevant solutions for cartilage repair and regeneration.

## 2. Materials and Methods

### 2.1. Formation of Decellularized Articular Cartilage Matrix

The human articular cartilage powder (hACM) was obtained from discarded distal femur allografts. Cartilage was removed from the articular surface with a 10-blade scalpel and subsequently flash frozen in liquid nitrogen. The frozen cartilage underwent multiple cycles of freezing at −80 °C for 24 h, lyophilization under a vacuum of 0.2 mBar at −50 °C for 24 h, and mechanical pulverization to a uniform particle size of less than 70 microns. This process was repeated until a homogenously fine powder was obtained, suitable for incorporation into the bioink. (Figure 1).

### 2.2. Synthesis of Experimental Bioink Formulations

The dry components of the experimental bioinks consisted of 20% weight per volume (w/v) medium viscosity alginate (A2033,Sigma-Aldrich, St. Louis, MO, USA), 5% w/v gum arabic (51198, Sigma-Aldrich, St. Louis, MO, USA), 5% w/v Polyvinyl alcohol (PVA) powder (P8136, Sigma, USA), and 5% w/v hACM powder. Separately, an established immortalized human articular chondrocyte line (C20A4, Millipore Sigma, Burlington, MA, USA) was expanded to confluence in standard culture media high-glucose DMEM with 10% FBS, 100 U/mL penicillin, and 100 µg/mL streptomycin. The confluent cells were trypsinized and suspended at a concentration of 5 × 10^6^ cells/mL. This cellular suspension was used to reconstitute the bioink at the appropriate w/v concentration.

### 2.3. Bioprinting Cell Laden Constructs

A bioprinter (BIO X™, Cellink, Gothenburg, Sweden) was used to print cylindrical hydrogel scaffolds using a 22-gauge needle. The constructs had a dimension of 10 mm D × 6 mm H and were printed with a concentric infill with 50% density using the above-described bioink. Immediately after printing, the cartilage constructs were cross-linked via submersion in 500 mM sterile calcium chloride for 30 min. The constructs were then rinsed 5 times with sterile tissue culture grade phosphate-buffered saline (PBS) prior to microscopy or culture. There were a total of 31 constructs printed for scanning electron microscopy (SEM) imaging (n = 1), and for cellular viability analysis (n = 5) and gene expression analysis (n = 5) at each of the three time points. One construct was imaged with SEM immediately after crosslinking and the remaining thirty constructs were cultured.

### 2.4. Culture of 3D Bioprinted Constructs

After printing, crosslinking, and rinsing, 30 of the cell-laden constructs were cultured in standard medium as described above. After 1, 7, and 14 days, a cohort of constructs (n = 10) underwent subsequent analysis.

### 2.5. Cell Survivability in Bioprinted Scaffolds

A commercial LIVE/DEAD Viability/Cytotoxicity Kit Kit (Invitrogen, Carlsbad, CA, USA) was used to evaluate the overall viability of chondrocytes within the composite hydrogels at 1, 7, and 14 days following printing. At each time point, five constructs were washed in PBS and allowed to incubate in a complete growth medium containing 2 µM calcein AM and 4 µM ethidium homodimer-1 for 30 min at 37 °C and 5% CO_2_, per manufacturer’s instructions. Following incubation, the constructs were rinsed in PBS and imaged under a fluorescent microscope (Leica Microsystems, Wetzlar, Germany). The fluorescence from these dyes was observed separately. The resulting live/dead images were analyzed with ImageJ software (NIH Image, Bethesda, MD, USA). Equation (1) was used to calculate percent viability.
(1)$$\mathrm{Viability}\;(\%)=\frac{{\mathrm{Intensity}}_{\mathrm{Calcein}\;\mathrm{AM}}}{{\mathrm{Intensity}}_{\mathrm{Calcein}\;\mathrm{AM}}+{\mathrm{Intensity}}_{\mathrm{Ethidium}\;\mathrm{Homodimer}-1}}\times 100$$


### 2.6. Real-Time PCR Analysis of Chondrogenic Gene Expression in Bioprinted Scaffolds

Real-time polymerase chain reaction (RT-qPCR) was performed to measure expression levels of SRY-box transcription factor 9 (*SOX9*) and type II collagen (*COL2A1*), relative to housekeeping gene glyceraldehyde-3-phosphate dehydrogenase (*Gapdh*). Cell lysate and RNA isolation was performed using the RNeasy minikit (Qiagen, Hilden, Germany) per manufacturer’s instructions. Cell lysates were collected at days 0, 7 and 14 and stored at −80 °C until completion of experiment. All RNA samples were analyzed simultaneously. The concentration and purity of the isolated RNA was assessed using NANODROP 2000UV-Vis spectrophotometer (Thermo Scientific, Waltham, MA, USA). RNA was reverse transcribed into cDNA using iScript cDNA Synthesis Kit (Bio-Rad Laboratories, Hercules, CA, USA). RT-qPCR was performed using SsoAdvanced™ Universal SYBR^®^ GreenSupermix (Bio-Rad Laboratories, Hercules, CA, USA) and CFX96™ Real-Time SystemThermal Cycler (Bio-Rad Laboratories, Hercules, CA, USA) according to manufacturer’s instructions with three technical replicates. The primers (Integrated DNA Technologies, Coralville, IA, USA) used in RT-qPCR reactions are shown in Table 1.

Primers were designed to span different exons to ensure no amplification of genomic DNA. The quantitative cycles (Cq) were measured using three samples from each group and time point to calculate means and standard deviations. Relative gene expression compared to reference gene, GAPDH, was calculated from quantitative cycles (Cq) by Equation (2).
Relative RNA expression = 2^−(ΔΔCt)^ = 2^−(ΔCt(target) − ΔCt(GAPDH))^(2)

### 2.7. Statistical Analysis

Data representation and statistical analysis was performed using GraphPad Prism (version 6.01) software. Statistical significance was determined by one-way ANOVA. *p* values less than 0.05 were considered significant. Data are presented as the mean ± standard error.

## 3. Results

The bioink facilitated uniform printing of cylindrical constructs, as indicated by Figure 2A. The consistency in printing underscores the bioink’s suitability for creating standardized scaffolds for tissue engineering. SEM revealed a porous structure in the 3D-printed constructs immediately after cross-linking (Figure 2B–D). This porous architecture is crucial for nutrient diffusion and cellular migration, mimicking the natural cartilage environment.

Live/dead staining was performed to assess cell viability post printing (day 1) and after one (day 7) or two (day 14) weeks in culture. The staining process was meticulous, ensuring accurate assessment of cell viability within the constructs. Representative images of live/dead fluorescently stained constructs can be seen in Figure 3. These images provide a visual representation of the cellular health within the bioink matrix over time.

Immediately post-printing, the cellular viability was 87.2%. This initial high viability rate is indicative of the chondrocytes’ ability to survive the process of incorporation into the bioink as well as the print stress. The viability decreased significantly to 76.4% at day 7 (*p* < 0.001), possibly due to environmental stress or nutrient limitations within the scaffold. However, it recovered to 85.9% at day 14 in culture (Figure 4A), suggesting adaptation or regeneration of the cells within the bioink environment.

Chondrogenic gene expression was quantified using RT-qPCR performed on the messenger RNA isolated from the chondrocytes within the hydrogels at day 1, 7 and 14. COL2A1 and SOX9 were assayed as they are widely used markers of a chondrogenic phenotype. The results of the gene expression analysis are shown in Figure 4B,C. The hydrogel-encapsulated cells showed a significant increase in transcriptional activity of chondrogenic genes COL2A1 (*p* = 0.008) and SOX9 (*p* = 0.021) at day 14 compared to day 1. These results indicate a significant enhancement in the chondrogenic differentiation of the cells encapsulated within the hydrogel over the 14-day period. This increase in gene expression is a promising indicator of the bioink’s potential to support chondrogenic differentiation, a key requirement for successful cartilage tissue engineering.

## 4. Discussion

One of the most difficult pathologies to treat within orthopedics is injury or the degradation of articular cartilage. Cartilage is a complex connective tissue consisting predominantly of proteoglycans and collagens, with very limited ability for regeneration and it therefore relies on biological supplementation or surgical intervention for repair [6,8]. Crucial in these repair techniques is fostering environments conducive to the development of hyaline cartilage over fibrocartilage. This is a rapidly growing field within orthopedics and represented a USD 2.8 Billion market cap in 2023 [37]. With increasing funding, exciting advances in 3D bioprinting have led to the development of cartilage scaffolds that contain living cells which can create the appropriate environment for cartilage regeneration [38]. For this process to be effective, it is crucial that the cartilage cells are able to adhere, grow and proliferate in order to successfully support the engineered structure. Tissue engineering, particularly bioprinting, offers a promising solution by constructing 3D scaffolds that imitate the complex architecture of native tissues. MSC-impregnated hydrogels are often used for both articular and fibro-cartilage tissue engineering [13].

The development of this novel alginate-based 3D bioink represents a significant advancement in the field of cartilage tissue engineering. Alginate, a naturally occurring anionic polysaccharide, has previously been used extensively in bioprinting due to its ease of crosslinking and biocompatibility [39]. In a previous study examining cell viability across several hydrogel types, it was reported that alginate hydrogel best supported the development and proliferation of hyaline cartilage-like tissue instead of a fibrocartilage-like tissue. This was evidenced by alginate hydrogels, in contrast to GelMA and BioINK™, supporting the proliferation of cell lines that were stained primarily for type II collagen rather than type I [13]. Not only that, but alginate naturally aids in the development of a round MSC shape which has been shown to support the development of a chondrogenic phenotype [40]. However, alone, alginate hydrogels lack biological properties to generate an ideal 3D scaffold, and thus require the addition of other biomaterials to recapitulate the native cartilage environment, such as chondrogenic peptides, growth factors, or cartilage cells themselves. The ability of bioprinted constructs to secrete a native cartilage extracellular matrix is considered one of the key criterion for successful cartilage regeneration [41]. The integration of human articular cartilage into the bioink formulation capitalizes on the inherent biological properties of cartilage to enhance the regenerative potential of the printed constructs, in line with the growing interest in using tissue-specific extracellular matrices to promote stem cell differentiation and mimic native tissue structures.

Our study aimed to assess the bioink’s printability and its ability to support chondrogenic properties in chondrocytes, building on previous similar work from groups such as Bandyopadhyay et al., who demonstrated that with their bioink of silk methacrylate, polyethylene glycol diacrylate and chondrocytes, the bioprinting technique was capable of both maintaining cell viability and showing strong mechanical properties [42]. We found that the bioink and encapsulated cells could endure the stresses of printing and could successfully produce a porous, cell-laden 3D scaffold. This composite demonstrated sustained cellular viability and enhanced chondrogenicity after 14 days, suggesting potential for cartilage regeneration.

This study’s findings are crucial, showing that the bioink can successfully create a porous structure essential for nutrient diffusion and waste removal. The decrease in cell viability observed at day 7 possibly indicates an initial adaptation phase for the chondrocytes within the new microenvironment. However, cellular viability returned to baseline by day 14, which provided evidence that the alginate-based 3D bioink could promote cellular proliferation, and the brief decline in cell viability does not point towards any suggestion of cytotoxicity. This promising recovery in viability is indicative of the bioink’s potential to support long-term cell survival. It also suggests that the constructs should be cultured for a minimum of two weeks in future experiments prior to undergoing evaluation of their properties, with longer-term culture likely to provide more insight into their efficacy.

Additionally, the alginate-based 3D constructs successfully enhanced chondrogenic differentiation, as seen with gene expression analysis. Transcriptional activity of both *COL2A1* and *SOX9* significantly increased by day 14; these gene markers imply enhanced chondrocyte differentiation and type II collagen proliferation. This positive effect of chondrogenesis may in part be a result of the suitable microenvironment generated by the alginate hydrogel, as previous studies have demonstrated [31,43,44]. Other reports have demonstrated that the use of cartilage-derived extracellular matrix in the bioink similarly had a positive effect upon chondrocyte proliferation and ECM production [45,46]. Our findings are similar to previous studies involving bioinks with animal-derived cartilage powder [47]. Wu et al. investigated a bioink made of alginate, porcine articular decellularized extracellular matrix (dECM) and stem cells from rabbit adipose tissue, and found an increase in cell aggregation and SOX9 expression [48]. Additionally, Beck et al. found an increase in the expression of *COL2A1*, *SOX9* and aggrecan in rat bone marrow MSCs when porcine knee dECM was added to their cell-laden bioink [49]. The combination of both alginate and decellularized human articular cartilage matrix used in this study not only supports the survival of chondrocytes but also actively promotes their differentiation towards a chondrogenic lineage.

Echoing principles utilized in bone tissue engineering, the success of demineralized bone matrix (DBM) in bone repair is paralleled in our study by the use of decellularized cartilage matrix in the bioink. This approach is instrumental for guiding cellular responses and ensuring the biofidelity of the engineered tissue. The insights and success gained from DBM in bone regeneration may inform future cartilage repair strategies, potentially enabling the construction of multi-tissue interfaces and more complex joint structures.

The potential clinical implications of these pursuits are significant in a field that is in need of more widely available and more affordable interventions. There are significant limitations in allograft availability, donor site morbidity of autograft, the need for multiple surgeries for autologous chondrocyte implantation (ACI), and questions of cell viability after storage [50]. Osteochondral allografts go through a rigorous matching process, and commonly, the waiting list for these grafts can be between 3 and 6 months. Donor site morbidity of 0–57% has been reported in the literature with the use of osteochondral autografts [51]. All currently available ACI techniques require two surgical procedures, increasing the risk of possible complications. While chondrocyte viability decreases with time starting at graft procurement, current screening requires a minimum of 14 days to perform the following serologic and microbiologic tests: HIV type 1 and 2 antibody, total antibody to hepatitis B core antigen, hepatitis B surface antigen, HTLV-1/HTLV-2 antibody, hepatitis C antibody, syphilis assay, and HCV and HIV1 with nucleic acid amplification tests [52]. 

A bioprinted cartilage alternative would allow for decreased surgical wait times, which becomes increasingly important from a financial perspective with lost time from work, but also when considering the risk of further injuries to other structures within the knee, including the menisci and ligaments. Not only is the timing surrounding surgery important, but the ability to customize the sizing and shapes of these grafts to best address these cartilage lesions while leaving as much healthy intact cartilage as possible is also important.

This study also suggests exciting possibilities for the development of personalized treatments. The use of 3D printing technology allows for the creation of patient-specific constructs, tailored to the individual’s anatomy and pathology. This could potentially improve the efficacy of treatment and reduce the risk of complications. However, this also requires the use of advanced imaging and modeling techniques to accurately capture the patient’s anatomy and translate it into a 3D-printable model.

This study also opens several exciting possibilities for future research. One such direction is the exploration of other biomaterials that can be integrated into the bioink to enhance its properties. While the use of decellularized human articular cartilage matrix has shown promising results, other materials could potentially enhance the bioink’s properties. For instance, the incorporation of growth factors or other bioactive molecules could further promote chondrocyte proliferation and differentiation.

It is important to recognize the limitations of this study, such as the in vitro conditions not fully replicating the in vivo environment of joint mechanics and immune interactions. The long-term stability and functionality of the constructs under physiological loading conditions remain to be tested. Future in vivo studies are essential to assess the mechanical properties, and durability of the constructs over extended periods. Furthermore, the immune response to the implanted constructs is another critical aspect that needs to be evaluated. The body’s immune system can mount a response against foreign materials, leading to inflammation and rejection of the implant. Therefore, it is crucial to assess the biocompatibility of the bioink and its potential to elicit an immune response.

Another important consideration is the regulatory landscape. The use of bioinks and 3D printing in tissue engineering is a relatively new field, and regulatory guidelines are still evolving. It is crucial to ensure that this technology is safe and effective before it can be widely adopted in clinical practice. This involves rigorous testing and validation, both in vitro and in vivo. Furthermore, the use of human articular cartilage in the bioink formulation raises additional regulatory and ethical considerations.

Overall, this study presents a promising alginate-based 3D bioink integrated with human articular cartilage, supporting survival and chondrogenic differentiation of encapsulated chondrocytes. While further research is necessary to optimize the bioink formulation and assess long-term functionality in vivo, the findings provide a solid foundation for future therapeutic applications in cartilage repair.

## 5. Conclusions

In conclusion, this study has demonstrated that an alginate-based bioink, integrated with decellularized articular cartilage matrix, can maintain chondrocyte viability and promote chondrogenic activity, as evidenced by the sustained cell viability rates post-printing, which remained high after 14 days in culture. The observed rise in chondrogenic gene expression, with significant increases in COL2A1 and SOX9 transcriptional activity at day 14, suggest that this bioink is a viable candidate for cartilage tissue engineering applications. Drawing on the analogy with demineralized bone matrix, which has established its efficacy in bone tissue engineering, our findings provide preliminary evidence that similar success could be anticipated for cartilage regeneration with this innovative bioink. While the in vitro results are promising, further investigation through in vivo studies is essential to evaluate the clinical potential of the bioink for repairing cartilage damage and restoring joint function.

## Figures and Tables

**Figure 1 bioengineering-11-00329-f001:**
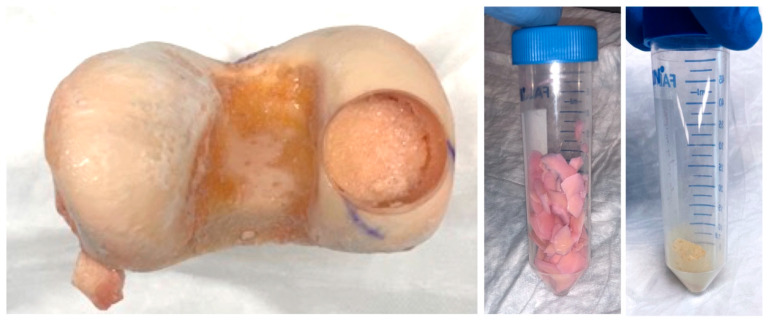
Articular cartilage preparation. Images shows cartilage harvest site on distal femur (**left**), resulting cartilage shavings (**middle**), and final powder used within the bioink (**right**).

**Figure 2 bioengineering-11-00329-f002:**
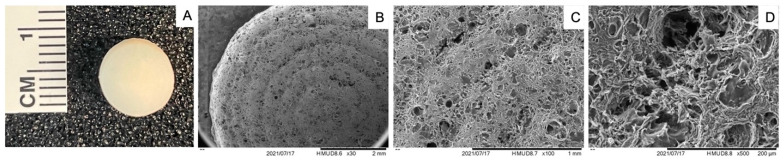
Visualization of 3D-printed constructs. (**A**) Gross image of biofabricated sample, (**B**–**D**) scanning electron microscopy images at 30×, 100×, and 500×, respectively.

**Figure 3 bioengineering-11-00329-f003:**
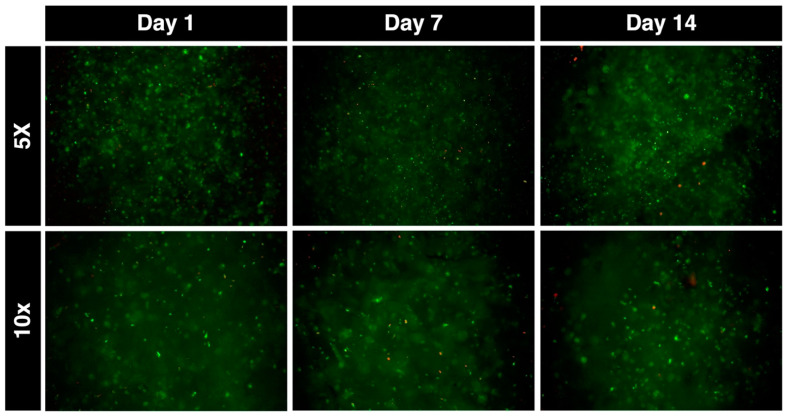
Fluorescence imaging. Fluorescence microscopy images of live (green) dead (red) staining of constructs at days 1, 7 and 14. Taken with 5× and 10× objectives.

**Figure 4 bioengineering-11-00329-f004:**
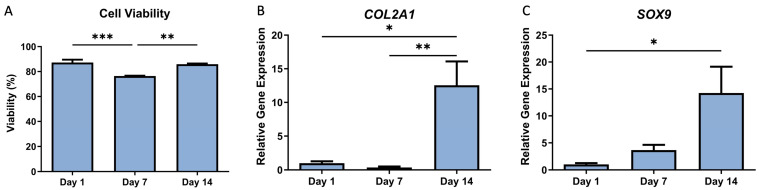
Cellular viability and gene expression of C20A4 cells. (**A**) Percent cell viability calculated by live dead staining intensity. (**B**) COL2A1 transcriptional activity, (**C**) SOX9 transcriptional activity. Error bars represent SD. * *p* < 0.05, ** *p* < 0.01, *** *p* < 0.001.

**Table 1 bioengineering-11-00329-t001:** Primer sequences used in RT-qPCR analysis. Displayed 5′–3′.

Gene	Forward Sequence	Reverse Sequence
Collagen type 2 alpha 1 (*COL2A1*)	CCCTGGTCTTGGTGGAG	CCATCATCACCAGGCTTTC
Sex-determining region Y-box 9 (*SOX9*)	TCTGAACGAGAGCGAGAA	GCGGCTGGTACTTGTAATC
Glyceraldehyde-3-phosphate dehydrogenase (*GAPDH*)	GTCTCCTCTGACTTCAACAGCG	ACCACCCTGTTGCTGTAGCCAA

## Data Availability

The data presented in this study are available on request from the corresponding author.

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
