# Peer review of "Fabrication of a Novel 3D Extrusion Bioink Containing Processed Human Articular Cartilage Matrix for Cartilage Tissue Engineering"

_bioengineering, 2024, doi:10.3390/bioengineering11040329_

Round 1

Reviewer 1 Report

Comments and Suggestions for Authors

The paper describes the use of alginate-based 3D bioink used to encapsulate human articular chondrocytes. The resulting 3D constructs were morphologically analyzed using SEM while fluorescence imaging was done to assess cell viability. The paper was well-written. Just one question: SEM micrographs as presented in Figure 2 showed porous structure of the 3D printed construct after crosslinking. It would be nice to show the progress of the crosslinking as previously described "After 1, 7, and 14 days, a cohort of constructs (n=10) underwent subsequent analysis." [line 129-130]. How will the morphology change after 1, 7 and 14 days if SEM analysis was performed alongside the results presented in Figure 3 (Fluorescence Imaging).

This is to recommend the paper for publication.

Author Response

Thank you for taking the time to review our manuscript and for recommending it for publication. Additionally, thank you for your important comment. While it would be relevant to analyze the porosity of our constructs at varying time points outlined, the ionic crosslinking step described herein takes place at a singular time point and thus the crosslinking cannot progress throughout the experimental timeline once removed from the calcium chloride solution. In the scope of this study, our primary focus was on the cellular and tissue-specific benefit of the cartilage additive. Now that we have established a foundation of potential benefits to utilizing this additive, we will take this comment into account and look to highlight material properties in subsequent experiments. You are absolutely correct in that we cannot rule out changes to our crosslinked hydrogel network (ie porosity, swelling and degradation) as the experiment progresses. We are currently undertaking follow-up studies to further characterize the rheological properties and degradation profiles of our proprietary bioink and the subsequent changes in these properties that may arise over the culture period.

Reviewer 2 Report

Comments and Suggestions for Authors

The manuscript titled “Fabrication of a Novel 3D Extrusion Bioink Containing Processed Human Articular Cartilage Matrix for Cartilage Tissue Engineering” introduces a novel alginate-based bioink infused with human decellularized articular cartilage matrix, aiming to assess its viability for a 3D bioprinting approach to cartilage tissue engineering. The topic is relevant and the manuscript deserves consideration. However, at presented state the manuscript needs revision to be considered further. The main comments and recommendations are listed below.

1. The authors carried out more qualitative than quantitative investigations, but, however, obtained some impact data. To show the importance of the work, it is better to reach the Abstract by more results (digit data) obtained during the experiment.

2.  L. 30-33. The recent studies show promising materials with anti-arthritic effect. Thus, the statement “Conventional treatments, often limited to joint fusion or replacement in late-stage degeneration are costly and provide only temporary symptom relief without restoring the original function of the tissue” is more practical, but does not reflect scientific side. It is recommended to make an overview of some promising materials that have potential anti-arthritic and then only provide justification of the perspectives and effectiveness of new types of cartilage scaffolds. For example, chicken embryo tissue hydrolyzate showed pronounced anti-arthritic effect https://doi.org/10.1002/fsn3.2529

3. L. 56-67. To show the deep understanding of the subject, it is recommended to expand overview and discussion on recent materials applied for tissue engineering as scaffolds. For example, bacterial cellulose https://doi.org/10.1016/j.ijbiomac.2023.128369 , chitosan https://doi.org/10.1016/j.ijbiomac.2017.05.067, etc.

4. L.102-103. Treatment parameters are missed.

5. L. 136-140. Such a detailed description of the reagents is impressive, but uninformative for most readers. The author could spell glycine as NH 2 ‐ CH 2 ‐ COOH as well, but, however, for sure the best options for readers are glycine, calcein AM, ethidium homodimer-1. If needed, the authors can give a reference with detailed spelling of chemicals.

6. Discussion can be reached by more references on the relevant recent studies with comparable results.

7. Conclusions should be supported by the results (data) obtained.

8. All abbreviations should be checked to be defined at the first use

Author Response

Thank you for taking the time to critically review our manuscript. Please see the attached document for detailed responses and changes made. We believe these changes have significantly improved our manuscript. 

Round 2

Reviewer 2 Report

Comments and Suggestions for Authors

The authors considered All comments and recommendations and decide them well. The revised manusript was improved and deserves consideration for publication.